# On the Interplay between Deadline-Constrained Traffic and the Number of Allowed Retransmissions in Random Access Networks

**DOI:** 10.3390/e26080655

**Published:** 2024-07-30

**Authors:** Nikolaos Nomikos, Themistoklis Charalambous, Risto Wichman, Yvonne-Anne Pignolet, Nikolaos Pappas

**Affiliations:** 1Department of Ports Management and Shipping, National and Kapodistrian University of Athens, 34400 Euboea, Greece; 2Department of Electrical and Computer Engineering, School of Engineering, University of Cyprus, 1678 Nicosia, Cyprus; charalambous.themistoklis@ucy.ac.cy; 3Department of Electrical Engineering and Automation, School of Electrical Engineering, Aalto University, 02150 Espoo, Finland; 4Department of Information and Communications Engineering, School of Electrical Engineering, Aalto University, 02150 Espoo, Finland; risto.wichman@aalto.fi; 5Dfinity Foundation, 6300 Zurich, Switzerland; yvonneanne@pignolet.ch; 6Department of Computer and Information Science, Linköping University, SE-60174 Linköping, Sweden; nikolaos.pappas@liu.se

**Keywords:** deadline-constrained traffic, packet deadlines, queuing, multi-packet reception, discrete-time Markov chains, delay-sensitive communications, low-latency communications

## Abstract

In this paper, a network comprising wireless devices equipped with buffers transmitting deadline-constrained data packets over a slotted-ALOHA random-access channel is studied. Although communication protocols facilitating retransmissions increase reliability, a packet awaiting transmission from the queue experiences delays. Thus, packets with time constraints might be dropped before being successfully transmitted, while at the same time causing the queue size of the buffer to increase. To understand the trade-off between reliability and delays that might lead to packet drops due to deadline-constrained bursty traffic with retransmissions, the scenario of a wireless network utilizing a slotted-ALOHA random-access channel is investigated. The main focus is to reveal the trade-off between the number of retransmissions and the packet deadline as a function of the arrival rate. Towards this end, analysis of the system is performed by means of discrete-time Markov chains. Two scenarios are studied: *(i)* the collision channel model (in which a receiver can decode only when a single packet is transmitted), and *(ii)* the case for which receivers have multi-packet reception capabilities. A performance evaluation for a user with different transmit probabilities and number of retransmissions is conducted. We are able to determine numerically the optimal probability of transmissions and the number of retransmissions, given the packet arrival rate and the packet deadline. Furthermore, we highlight the impact of transmit probability and the number of retransmissions on the average drop rate and throughput.

## 1. Introduction

Future wireless communication networks are envisioned to play a major role in enabling autonomous systems in the context of the Internet of Things (IoT), comprising connected vehicles, smart devices, or fully automated factories; see, for example, [1,2,3,4]. The data traffic produced from these wireless devices, referred to as machine-to-machine (M2M) communication, will significantly differ from the wireless traffic served by currently deployed wireless networks. In greater detail, wireless devices might transmit packets consisting of a few bytes of information, while being sporadically active. Moreover, a massive number of devices may demand ubiquitous connectivity, and the transmission of packets with extremely stringent latency and reliability requirements, as is the case of mission-critical M2M applications, supporting real-time closed-loop control, one of the essential mechanisms enabling such emerging applications [5,6,7].

### 1.1. Related Work

The rapid blooming of applications requiring deadline-constrained packet transmissions and multimedia broadcasting over wireless communication networks has stimulated research on deadline-constrained broadcasting, relying on scheduling [8,9,10,11] and random access [12,13,14,15,16,17,18,19,20]. The work in [12] obtained the optimal access probability of secondary nodes in a cognitive radio network, towards maximizing the successful delivery probability (SDP) under specific deadline constraints using simple slotted ALOHA. The issue of improving the reliability of deadline-constrained one-hop broadcasting, based on the slotted-ALOHA with retransmission was investigated in [13]. More specifically, the SDP was derived, as well as the optimal access probability for SDP maximization. Regarding retransmissions, their optimal value under specific throughput requirements was determined. Queuing analysis of deadline-constrained broadcasting, but without retransmission, was investigated in [14]. By modeling the system as a discrete-time Geo/Geo/1 queue with a specific delivery deadline, several performance metrics were investigated, including the loss probability, queue length distribution, mean waiting time, and SDP. Nevertheless, deadline-constrained broadcasting with retransmissions has not been analyzed yet. Furthermore, the paper in [15] studied a slotted-ALOHA network consisting of nodes with energy harvesting capabilities. For this setup, the author proposed an approximate analytical model for deriving the timely-delivery ratio, since the interaction of energy and data queues poses significant difficulties in obtaining an exact analytical model. In [16], the author derived the upper and lower throughput bounds in random access networks, employing power-domain non-orthogonal multiple access (NOMA). These expressions allowed for the acquisition of the traffic intensity, maximizing the lower bound, while the asymptotic maximum throughput was obtained for a large number of power levels. However, the impact of retransmissions and deadlines on the throughput performance was not investigated. Then, the author in [17] adopted the exploration of multi-channel ALOHA through preambles before transmitting data packets in machine-type communication (MTC), showing a maximum throughput improvement by a factor of $2-e^{-1}\approx 1.632$. In addition, a steady-state analysis with fast retrial was performed, highlighting that the delay outage probability is significantly reduced for a lightly loaded system. Another work examined a slotted-ALOHA random access network in [18], focusing on the age of information (AoI) performance. In greater detail, a stationary threshold-based age-dependent random access (ADRA) protocol was presented, enabling transmitting nodes to access the channel with a certain probability, as long as their instantaneous AoI surpassed a predefined threshold. Simulation results showed that ADRA provided better AoI performance than other age-oriented random access protocols. It should be noted that the impact of retransmissions and message expiration was not incorporated in the results. A novel random medium access control (MAC) scheme was presented in [19], relying on transmission opportunity prediction, measuring the transmitted-side interference. The authors used stochastic geometry to model the network topology and obtained a fixed-point equation to provide the optimal transmission probability to maximize the proportional fair throughput in the network. Even though the proposed scheme showed improved throughput performance, allowing more nodes to access the wireless medium, it introduced additional complexity by implementing a transmission opportunity prediction mechanism. Focusing on deadline-constrained broadcasting in random access networks, the authors in [20] aimed at enhancing the maximum achievable timely delivery ratio. Their solution relied on dynamic control, allowing each active node to accurately determine the transmission probability by considering the current delivery urgency and the knowledge of current contention intensity. Moreover, optimal control policies for cases where the contention intensity is fully available or incompletely known were derived, applying the Markov decision process (MDP) theory. Nevertheless, the presented results relied on the assumption of i.i.d. traffic, without considering generalized traffic patterns or bursty traffic.

### 1.2. Contributions

Overall, the contributions of this paper are as follows:
First, the successful transmission probability is obtained for the case where a new packet is generated after the successful transmission of the previous one has been completed or where that packet has been dropped, either due to reaching the maximum number of allowed retransmissions or expiration. In this part of the analysis, no data buffering is considered.The second part of the analysis investigates stochastic bursty traffic with buffer-aided devices by means of a discrete-time Markov chain (DTMC) for single- and multi-packet reception. Two cases are distinguished:–First, the case where a packet keeps being retransmitted until expiration or successful transmission. This case resembles that of [14]; however, a different DTMC is constructed, providing the SDP directly;–Second, the case where the deadline value is larger than the number of allowed retransmissions. A similar construction of the DTMC is used and, as a result, the system performance is analyzed.Simulation results are included, validating the theoretical findings and demonstrating the effect of different transmit probabilities and numbers of allowed retransmissions on the drop rate and the average throughput. Finally, the positive effect of MPR in the network, as expected, is highlighted.

It should be noted that, in this work, we focus on a single-carrier topology where two cases of packet reception by the destination are investigated, i.e., single- and multi-packet reception. Thus, in the results, we assume that multiple nodes are concurrently served on the same carrier. When a multi-carrier network is considered, our analysis and findings can be easily extended by separately investigating the reception performance for each carrier.

The findings of this study may be useful in practical code domain-based multi-packet reception (MPR), such as code-division multiple access (CDMA) and sparse code division multiple access (SCMA) systems, supporting deadline-constrained applications. At the same time, this paper can serve as a building block for investigating packet scheduling in random access cooperative networks with deadline constraints.

### 1.3. Structure

The remainder of the paper is structured as follows: Section 2 presents the system model adopted in this study and the necessary preliminaries. Then, Section 3.1 and Section 3.2 present the theoretical framework and analysis for the SDP when a packet is generated after the previous one has either been successfully transmitted or dropped. Furthermore, the impact of bursty traffic and buffering is also analyzed. In Section 4, the numerical and simulation results are presented, giving several insights about the performance of the considered scenario. Finally, in Section 5, we draw conclusions and discuss possible extensions and future research directions.

## 2. System Model and Preliminaries

In this section, the system model and the necessary preliminaries for the development of our study are presented.

### 2.1. Notation

Table 1 includes the notation used throughout the paper.

### 2.2. Network Model

In this work, a network comprising *N* nodes, being in transmission range and sharing the same wireless channel, is considered. While we could consider the case in which each node *i* in the network has its intended destination, in this paper, without loss of generality, we concentrate on the case where all the nodes transmit towards a common destination; this also motivates our work on multi-packet reception. Random access of the wireless medium is assumed and, thus, each node transmits with probability $q_{i}$ (for simplicity of exposition, the same value of $q_{i}$ is assumed for all nodes, i.e., $q_{i}=q\;\forall i$). The time is slotted and each packet transmission requires one slot.

A widely used protocol to maintain reliable packet transmissions over unreliable communication is the automatic repeat request (ARQ) ([21], §5). Herein, we assume that nodes invoke ARQ, in which instantaneous and error-free acknowledgements/negative-acknowledgements (ACK/NACK) are transmitted by the receiver over a separate narrow-band channel.

Regarding the packet deadlines, when a new packet *j* is generated, it has a deadline $D_{j}$ (for simplicity, it is assumed that $D_{j}$ is the same for all packets, i.e., $D_{j}=D\;\forall j$) and immediately enters the queue (in the case of bursty traffic). If the packet deadline expires before the packet reaches the destination, then the packet is dropped from the queue. Here, two cases are considered for the number of allowed retransmissions:
(a)the case where each packet can be retransmitted until its expiration (i.e., retransmitted up to D−1 times), and(b)the case where the number of retransmissions is equal to *n*, where 1≤n<D−1.

Retransmissions are necessary when the packet does not reach the destination, either due to a collision or unsuccessful packet reception at the receiver, as a result of the wireless nature of the channel. As a convention, the transmission of a packet from the queue is performed at the start of the slot and packets enter the queue at the end of the slot.

### 2.3. Physical Layer Model

Let $P_{i}$ be the transmit power of node *i*, and $\sqrt{g_{i}}$ be the channel coefficient between node *i* and the receiver. The received power, when node *i* transmits, is therefore $g_{i}s_{i}$, where $\sqrt{g_{i}}$ is a random variable (RV) representing small-scale fading and $s_{i}$ is the received power factor. The received power factor $s_{i}$ is given by $s_{i}=P_{i}r_{i}^{-\alpha }$, where α is the path loss (PL) exponent. Let $T_{i}$ denote the set of nodes whose transmitted power appears as interference to the transmitted signal from node *i*. If node $j\in T_{i}$, this transmission will appear as interference to the receiver when decoding the message from node *i* with a power of $g_{j}s_{j}$. The total interference and noise, denoted by $I_{i}$, at the receiver when serving node *i* is given by
(1)$$\begin{array}{c} I_{i}=\sum_{j\in T_{i}}g_{j}s_{j}+\eta , \end{array}$$
where η≥0 denotes the power of noise at the receiver other than interference from other nodes.

It is considered that a packet from node *i* is successfully transmitted to the receiver if, and only if, the signal-to-interference-and-noise ratio (SINR) exceeds a certain threshold γ, called the capture ratio, i.e.,
(2)$$\begin{array}{c} {\mathrm{SIN}R}_{i}=\frac{g_{i}s_{i}}{\sum_{j\in T_{i}}g_{j}s_{j}+\eta }\geq \gamma . \end{array}$$
When only node *i* transmits, the packet is successfully transmitted to the receiver if, and only if, the signal-to-noise ratio (SNR) exceeds γ, i.e.,
(3)$$\begin{array}{c} {\mathrm{SNR}}_{i}=\frac{g_{i}s_{i}}{\eta }\geq \gamma . \end{array}$$

Under Rayleigh fading, $g_{i}$ is exponentially distributed [22]. When only node *i* transmits, the success transmission probability for node *i* is given by
(4)$$\begin{array}{c} p_{i,0}=P\left[g_{i}\geq \frac{\eta \gamma }{s_{i}}\right]=\exp\left(-\frac{\eta \gamma }{v_{i}s_{i}}\right), \end{array}$$
where $v_{i}$ denotes the Rayleigh fading RV parameter (i.e., $g_{i}∼\mathrm{Rayleigh}\left(v_{i}\right)$).

When $|T_{i}|$ nodes transmit simultaneously with node *i*, the successful transmission probability for node *i* is given by ([23], Theorem 1)
(5)$$\begin{array}{c} p_{i,T_{i}}=\frac{\exp\left(-\frac{\gamma \eta }{v_{i}s_{i}}\right)}{\prod_{k\in T_{i}}\left(1+\gamma \frac{v_{k}s_{k}}{v_{i}s_{i}}\right)}. \end{array}$$

### 2.4. Preliminary Analysis

The packet arrival process at a node *i* is assumed to follow the Bernoulli distribution with an average probability λ. Let μ denote the probability that a packet at the head of the queue of node *i* will be successfully transmitted to the destination in a given time slot. Then, for the case when b−1 other nodes are backlogged (where b≤N), and only node *i* transmits, μ is given by $\mu_{1}$, where $\mu_{1}$ is
(6)$$\begin{array}{c} \mu_{1}=p_{i,0}q{\left(1-q\right)}^{b-1}. \end{array}$$
When b−1 other nodes are backlogged, and apart from node *i*, c−1 specific other nodes transmit (2≤c≤b), then μ is given by $\mu_{c}^{1}$, where $\mu_{c}^{1}$ is
(7)$$\begin{array}{c} \mu_{c}^{1}=p_{i,c-1}q^{c}{\left(1-q\right)}^{b-c}. \end{array}$$
If we consider the case of any c−1 nodes out of the b−1 other nodes to transmit, then μ is given by $\mu_{c}$, where $\mu_{c}$ is
(8)μc=b−1c−1pi,c−1qc(1−q)b−c.
Combining all these cases together, the probability μ that a packet at the head of the queue of node *i* will be successfully transmitted to the destination in a given time slot is
(9)μ:=∑k=1bμk=∑k=1bb−1k−1pi,k−1qk(1−q)b−k,
where *k* denotes the number of nodes transmitting simultaneously, including node *i*.

## 3. Theoretical Analysis

This section provides a theoretical analysis for the SDP and the effect of bursty traffic and buffering.

### 3.1. SDP

In this part, the SDP analysis of a packet being at the head of the queue is given. Here, a new packet is generated after the previous one has been successfully delivered or dropped. Note that a packet might be dropped either due to expiration or by exceeding the number of allowed retransmissions *n*. Two cases are considered: (1) when n=0 (i.e., one transmission is allowed), and (2) when 0<n≤D−1 (the number of total transmissions is more than one but less than or equal to the deadline of the packet).

When the traffic is characterized by deadlines, an important metric is the SDP, given that the packet is at the head of the queue, $p_{s}\left(n,D\right)$, which is the probability that a packet will be successfully delivered to the destination prior to its expiration or surpassing the total number of allowed transmission attempts, m≡n+1.

#### 3.1.1. n=0

First, the case for which no retransmissions are allowed is considered, i.e., when the allowed number of transmission attempts, *m*, is m=1. Then, $p_{s}\left(n,D\right)=p_{s}\left(0,D\right)$ and becomes
(10)$$\begin{array}{cc} p_{s}\left(0,D\right) & =\mu +\left(1-q\right)\mu +\ldots +{\left(1-q\right)}^{D-1}\mu \\ \; & =\mu \sum_{k=1}^{D}{\left(1-q\right)}^{k-1}=\frac{\mu [1-{\left(1-q\right)}^{D}]}{q}. \end{array}$$
Let ν≜μ/q, then (Equation 10) becomes
(11)$$\begin{array}{c} p_{s}\left(0,D\right)=\nu \left[1-{\left(1-q\right)}^{D}\right]. \end{array}$$

#### 3.1.2. 0<n≤D−1

Let *S* denote the event that a packet will be successfully transmitted within the desired deadline, and let *X* denote the event of the first transmission attempt. Then, $p_{s}\left(n,D\right)$, 1≤n≤D, is given by [13]
(12)$$\begin{array}{c} p_{s}\left(n,D\right)=\sum_{k=1}^{D}P\left(S,n|X=k\right)P\left(X=k\right), \end{array}$$
where P(·) denotes the probability of an event and P(·|·) denotes the conditional probability of an event. Hence, the value of $p_{s}\left(m,D\right)$ can be computed iteratively by
(13)$$\begin{array}{cc} p_{s}\left(n,D\right)= & \;q\left[\nu +\left(1-\nu \right)p_{s}\left(n-1,D-1\right)\right]\\ \; & +\left(1-q\right)q\left[\nu +\left(1-\nu \right)p_{s}\left(n-1,D-2\right)\right]\\ \; & +{\left(1-q\right)}^{2}q\left[\nu +\left(1-\nu \right)p_{s}\left(n-1,D-3\right)\right]\\ \; & +\ldots +{\left(1-q\right)}^{D-1}q\left[\nu +\left(1-\nu \right)p_{s}\left(n-1,0\right)\right]\\ = & \;\sum_{k=1}^{D}{\left(1-q\right)}^{k-1}q\left[\nu +\left(1-\nu \right)p_{s}\left(n-1,D-k\right)\right]. \end{array}$$

### 3.2. The Effect of Bursty Traffic and Buffering

In the previous section, we studied the case that a packet is directly generated once the previous one has been successfully transmitted or dropped. In this section, we consider the impact of bursty traffic and data buffering on the system performance. In greater detail, in a time slot, a packet will be generated with a given deadline, according to a probability λ, and will enter the queue. Our analysis focuses on a single node having bursty traffic, while the rest of the N−1 nodes are considered to always be backlogged, i.e., at least one packet always resides in the queues. It should be noted that this is the worst-case scenario, as the number of transmissions, and consequently the number of collisions, in a time slot are overestimated. A DTMC model of a system with bursty traffic is used, while the arriving packets are stored in a queue. Based on the number of allowed retransmissions, we consider two cases: (1) each packet can be retransmitted as many times as required for as long as it remains in the queue (i.e., n=D−1 number of retransmissions); (2) a limited number of retransmissions (i.e., n<D−1) is allowed for each packet.

#### 3.2.1. Number of Retransmissions n=D−1

Here, the number of possible retransmissions is equal to the deadline. For a given deadline *D*, the DTMC has D+1 states, where the states from 0 to *D* model the time passed. The DTMC includes two additional *virtual* states (i.e., states *S* and *U*) capturing the events of successful and unsuccessful delivery, respectively. State *S* is included to capture the successful packet transmission before expiration or dropping. The values of the transition probabilities from state *S* to any other state depend on the state prior to *S*. Then, state *U* models the event where a packet either expires or is dropped. An illustrative example of the DTMC for a deadline D=3 is shown in Figure 1.

The transition probability matrix *M* for a deadline of D=3, which is column stochastic by construction, is given by
(14)$$\begin{array}{c} M=\left[\begin{array}{cccc} 1-\lambda & \mu (1-\lambda ) & \mu {\left(1-\lambda \right)}^{2} & \mu {\left(1-\lambda \right)}^{3}\\ \lambda & \mu \lambda & \mu \lambda (1-\lambda ) & \mu \lambda {\left(1-\lambda \right)}^{2}\\ 0 & 1-\mu & \mu \lambda & \mu \lambda (1-\lambda )\\ 0 & 0 & 1-\mu & \mu \lambda \end{array}\right]. \end{array}$$

#### 3.2.2. Number of Retransmissions n<D−1

In this case, a packet drop might occur for two reasons: either the number of retransmissions has reached its maximum allowed value or the packet has expired. The main motive behind limiting the number of retransmissions is to reduce the number of packets residing in the queue and waiting to be transmitted, thus reducing the amount of packets that will expire, and increasing the system throughput. Given a deadline *D*, the DTMC has
(15)$$\begin{array}{c} D\left(n+1\right)+1-\frac{n(n+1)}{2} \end{array}$$
states including, similarly to the previous case, two additional *virtual* states, facilitating the visualization of the successful transmissions. For the case in which one retransmission is allowed (n=1), for example, the DTMC has 2D states (for a deadline of 3, as illustrated in Figure 2, 6 states exist). As can be seen in (Equation 15), the construction complexity of the DTMC is only determined by the deadline *D* and the number of maximum retransmissions *n* and, hence, is independent of the number of users. The number of users only affects the transition probabilities of the DTMC. Hence, the results presented in this paper can also be applied to massive multi-access networks.

**Remark** **1.**
*Constructing the DTMCs that include these virtual states offers a framework for investigating the behavior of such advanced wireless communication systems, where packets may have a limited number of allowed retransmissions.*


The throughput, *T*, of the system can be obtained by considering the probability of being in the virtual state *s*. This can be found by identifying the states from which a successful transmission can occur. In greater detail, the throughput is given by
(16)$$\begin{array}{c} T=\sum_{s\in S}\pi \left(s\right)\mu , \end{array}$$
where π(s) denotes the steady state of state *s* in the DTMC, *s* being a state of the DTMC belonging to the set of states S capturing the event of a successful transmission. In the case of the DTMC in Figure 2, the set S consists of the states {1,(2,0),(2,1),(3,0),(3,1)}.

The drop rate (or drop probability), DR, is correspondingly derived by considering the states from which a packet might be dropped, due to violating its deadline or absence of retransmissions. More specifically, the drop rate is given by
(17)$$\begin{array}{c} DR=\sum_{f_{D}\in F_{D}}\pi \left(f_{D}\right)\left(1-\mu \right)+\sum_{f\in F}\pi \left(f\right)\left(q-\mu \right), \end{array}$$
where $\pi (f_{D})$ is the steady state of the states $f_{D}$ in the DTMC, belonging to the set of states $F_{D}$, whose transmission is last. In the case of the DTMC in Figure 2, $F_{D}$ consists of the states {(3,0),(3,1)}. The remaining states from which the last unsuccessful transmission of a packet can take place belong to the set of states F. In the case of the DTMC in Figure 2, F consists of the state {(2,1)}.

**Remark** **2.**
*In order to find the optimal transmission probability q with respect to a certain metric (e.g., minimize DR or maximize T), an optimization problem can be formulated. For example, to maximize throughput T, we propose the following optimization problem:*

(18)
$$\begin{array}{c} \max_{q}\sum_{s\in S}\pi \left(s\right)\mu . \end{array}$$

*Note that both μ and the steady-state distribution π are functions of q, but since there is no known analytical expression of π with regards to q (yet), it is not possible to provide an analytical solution; hence, to solve this problem, we resort to numerical approaches.*


## 4. Simulation and Numerical Results

This part presents simulation and numerical results using MATLAB^®^—R2022a for a topology comprising buffer-aided nodes with buffer size L=3 and varying packet deadline values *D*. Our analysis does not consider the impact of varying the buffer size, as we assume that the buffer size is at least as big as the packet deadline value, and in cases where the buffer size exceeds this value, the packets at the back of the queue will never be transmitted. The topology is depicted in Figure 3, and it is considered that all the transmitting nodes have the same distance to the common destination. More specifically, the performance of a non-backlogged user is evaluated, in terms of drop rate and average throughput, measured in bps/Hz, for 10^5^ time-slots, various transmit probabilities *q* and numbers of retransmission *n* values. It is noted that various cases are examined regarding the number of users in the network, i.e., N=2,3,4,5, where apart from the non-backlogged user, the other users are assumed to be backlogged. In addition, for the non-backlogged user, the packet arrival probability is λ=0.5, unless otherwise stated and the probability of a successful transmission is $p_{i,0}=0.75$ for the scenario with the collision channel model, while when MPR is allowed $p_{i,1}=0.375$, $p_{i,2}=0.1875$, $p_{i,3}=0.09375$ and $p_{i,4}=0.046875$, calculated from (Equation 4) and (Equation 5) using the parameters in Table 2. Thus, here the MPR assumes that the receiver is equipped with multiple single-user detectors. It is noted that the network parameters in Table 2 were selected to correspond to an indoor environment where, due to clutter, line-of-sight (LoS) is not guaranteed and low-power nodes, such as sensors, transmit their data to a common sink, similarly to in the case of an Industry 4.0 topology.

### 4.1. Collision Channel Model

The first scenario focuses on a topology with N=2 users, allowing only single-user transmission. So, when both users aim to access the channel, a collision occurs.

#### 4.1.1. Number of Retransmissions n=D−1

The impact of varying *q* for n=D−1=2 allowed retransmissions is considered in this scenario. Figure 4 shows the drop rate results for different values of transmit probability *q*, equal for the two users. It can be seen that the drop rate for the non-backlogged user is minimized for a transmit probability value q=0.5. For lower *q* values, an increase in the drop rate is experienced, as the device does not access the frequently vacant wireless channel. In these cases, packets reside for more time slots in the queue, being consequently dropped due to expiration. Moreover, for higher *q* values, collisions may occur and the drop rate increases. Regarding the effect of λ, increasing the packet arrival probability results in an increased drop rate.

Then, Figure 5 illustrates the average throughput performance for different *q* values. Here, the maximum throughput is obtained when q=0.5. As has been already observed for the drop rate results, when this *q* value is adopted, the non-backlogged user enjoys improved throughput, as more packets are successfully transmitted towards the destination. More specifically, a q=0.5 optimizes the trade-off between channel access and collisions with the packets of the backlogged user. In addition, increasing λ results in increased throughput for the non-backlogged user, while saturation can be seen when varying λ=0.5 to λ=0.75 and more evidently for λ=0.95.

#### 4.1.2. Number of Retransmissions n<D−1

For this scenario, the maximum number of allowed retransmissions in the network is n=1. Figure 6 shows the drop rate performance for different *q* values. Again, a similar trend to the first scenario is observed. More specifically, for the non-backlogged user, the minimum drop rate is seen when q=0.5. As for the effect of *n* on the drop rate, a negligible increase is observed for larger *q* values, since *n* is equal to one and packets are dropped more often from the system, after a collision. It must be noted that similarly to the previous case, higher λ values lead to more packets being dropped.

Finally, Figure 7 depicts average throughput results for varying *q* values. The throughput performance closely follows that of the first scenario. So, for q=0.5, the maximum throughput is acquired for the non-backlogged user, while for higher *q* values, the average throughput slightly degrades. In conclusion, when both cases of *n* are considered, it can be observed that the performance of the non-backlogged user does not change and the number of retransmissions marginally affects the drop rate and the average throughput given small deadline values and a low number of users. Finally, throughput performance improves when higher λ values characterize the non-backlogged user’s traffic, resulting in saturation.

#### 4.1.3. Comparisons

In this case, the system performance is compared, when all the packets are characterized by deadline values D=5, and additionally, *(i)* no retransmissions are attempted (n=0), *(ii)* some retransmissions are attempted (in this case n=1), and *(iii)* as many retransmissions as needed are allowed (n=D−1=4). We see in both Figure 8 and Figure 9 that the performance is improved, as more retransmission attempts are allowed. This is expected as, in this case, there are no packets with the same or lower deadlines residing further back in the queue, due to the fact that the inflow of packets is low and an equal deadline is assumed for all the packets.

### 4.2. Multi-Packet Reception

The second scenario investigates the impact of allowing MPR on the drop rate and throughput performance of the network. Here, the number of users *N* varies, i.e., N=2 or N=3 and D=3.

#### 4.2.1. Number of Retransmissions n=D−1

Figure 10 depicts the drop rate performance when n=2 retransmissions are allowed in a network with N=2 users. As both users simultaneously transmit, a significantly different system behavior is observed compared to the previous scenario, due to the MPR capability. In greater detail, considering the fixed value of the successful transmission probability *P*, it can be observed that the drop rate reduces, as *q* increases and MPR is allowed, i.e., collisions do not occur in the network. In addition, increasing λ results in more instances of dropped packets, regardless of *q*.

Next, Figure 11 shows the throughput performance of the network. As was observed for the drop rate, the average throughput performance improves when the value of *q* increases. Since MPR is allowed, allowing both users to access the medium more frequently has a beneficial effect on the number of transmitted packets, and thus the average throughput is increased. In this case, a higher λ corresponds to significantly higher throughput as *q* increases, while for lower *q* values, the throughput is slightly affected. However, it is clearly seen that throughput performance saturates after λ=0.75.

#### 4.2.2. Number of Retransmissions n<D−1

Figure 12 illustrates the drop rate performance for the same network when n=1 retransmission is possible. It can be seen that the reduction in allowed retransmissions results in an almost negligible drop rate performance degradation for the non-backlogged user. Overall, the drop rate reduces in an identical fashion as in the case of n=2 retransmissions, as *q* increases. On the contrary, when λ increases, the drop rate performance degrades.

Likewise, the average throughput performance is largely unaffected by reducing the value of *n*, as can be seen in Figure 13. More specifically, the average throughput sees a noteworthy increase at around 7% for q=0.9, i.e., when both users try to transmit with high probability. In this comparison, when λ=0.25, a slightly improved throughput is observed compared to the corresponding case of n=2, but for the other two λ values, the throughput is almost negligibly reduced.

#### 4.2.3. Comparisons

Here, different cases with N=2,3,4,5 users are considered and the impact of varying *c*, i.e., the number of simultaneously transmitting users on the drop rate and throughput performance is examined.

The drop rate results are depicted in Figure 14. It can be seen that when c=1, the drop rate first reduces until q=0.3 for N=2,3 and until q=0.2 for N=5, since packet expiration due to inactivity is avoided. Then, as *c* increases, a worse drop rate performance is observed for all *N* values. However, when all users simultaneously transmit (N=c) and MPR is performed, the drop rate performance is enhanced, as *q* increases, independently of the number of users in the network. Meanwhile, in this case, the drop rate increases as *N* increases, due to the higher levels of interference in the network.

Finally, Figure 15 shows the average throughput results for the three different cases of *N* and a varying number of concurrently transmitting users *c*. Again, the results indicate that when MPR is allowed, the throughput is improved when all the users are able to transmit and access the channel with a higher probability. Moreover, increasing the number of users in the network *N* negatively affects throughput, as interference arises. In addition, for the case of c=1, a q>0.3 for N=3,4 and a q>0.2 for N=5 leads to reduced throughput.

It should be noted that even though we investigated the network performance for a different number of users *N*, taking values in the set N={2,⋯,5}, these users share a single wireless channel. Thus, the results presented in this study can facilitate the optimization of the trade-off between spectral efficiency and the number of concurrently transmitting users, considering their corresponding SDP and the service requirements. In addition, in practical network deployments, as the available spectral resources increase, the number of coexisting users and machines on multiple channels will correspond to the massive connectivity cases of IoT and MTC scenarios.

## 5. Summary and Future Directions

### 5.1. Summary

A network consisting of wireless devices transmitting deadline-constrained data using slotted-ALOHA random-access channels was considered. The performance of deadline-constrained transmission with retransmissions was studied, as the transmission of packets residing in queues might experience delays from retransmissions and they might become dropped before their turn comes for transmission. The goal of this study was to reveal and investigate the trade-off between the packet deadline and the number of allowed retransmissions, as a function of the packet arrival rate. Towards this end, the system was analyzed by using Markov chains, while numerical and simulation results highlighted the impact of different transmit probability values and the number of allowed retransmissions on the drop rate and the average throughput. Moreover, the effect of allowing multi-packet reception for different numbers of simultaneously transmitting users was shown.

### 5.2. Future Directions

There are several interesting research directions for extending this paper, in the context of MTC and 6G networks. One aspect that should be studied is related to MPR performance with direct sequence CDMA (DS-CDMA) or SCMA for improved multiple access by allowing multiple simultaneous transmissions [24]. In addition, the case of random access in cooperative networks with deadline constraints, departing from orthogonal schemes based on packet scheduling [25] should be analyzed. Moreover, the performance of random access in the context of grant-free NOMA [26,27] represents another important research area, together with the development of analytical methods to optimize the transmission probability *q* for each scenario. Furthermore, other metrics must be considered in the analysis, such as the freshness of the data [28,29]. Additional directions include the deployment of hybrid ARQ (HARQ) mechanisms, which encode the data with a forward error correction code, thus making the SDP higher in every retransmission [30,31], as well as power control mechanisms, studying the interplay between improving the successful transmission probability $p_{i,0}$ and the number of transmissions [32]. Finally, the adoption of machine learning to predict the transmission patterns of concurrently transmitting nodes through privacy-preserving federated learning algorithms [33] may be beneficial for network performance.

## Figures and Tables

**Figure 1 entropy-26-00655-f001:**
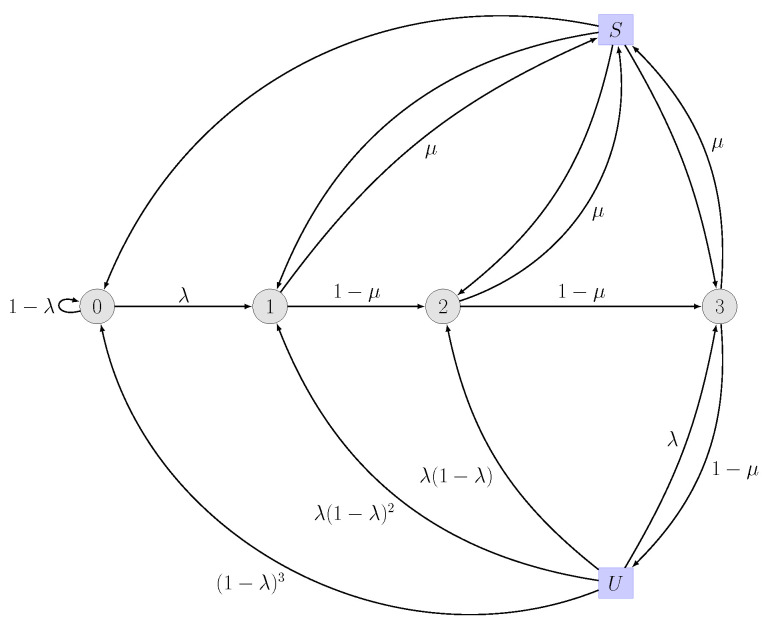
DTMC with D=3 when the number of allowed retransmissions is n=2. Note that the number of states in this case (n=D−1) is equal to D+1 states, including two additional virtual states capturing the events of successful (state *S*) and unsuccessful (state *U*) delivery.

**Figure 2 entropy-26-00655-f002:**
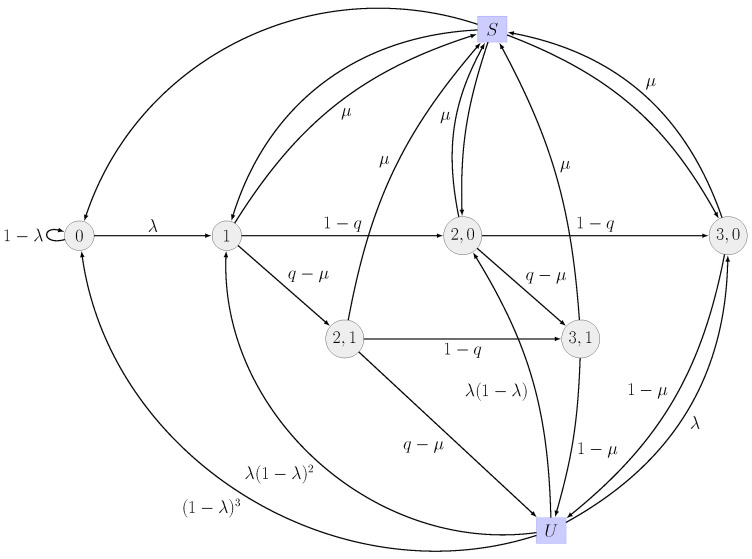
DTMC with D=3 when the number of allowed retransmissions is n=1. Note that the number of states in this case (n<D−1) is as given in (Equation 15). The idea behind this construction is to capture the state of all the packets in the queue.

**Figure 3 entropy-26-00655-f003:**
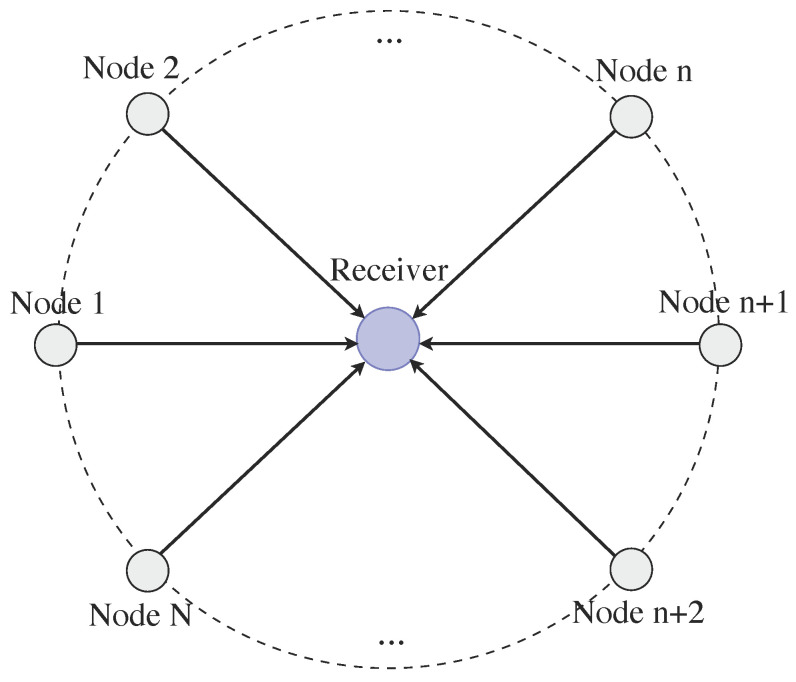
The considered network topology with *N* transmitting nodes and a common destination. The network is chosen in this topology such that all the transmitting nodes have the same distance towards the common destination.

**Figure 4 entropy-26-00655-f004:**
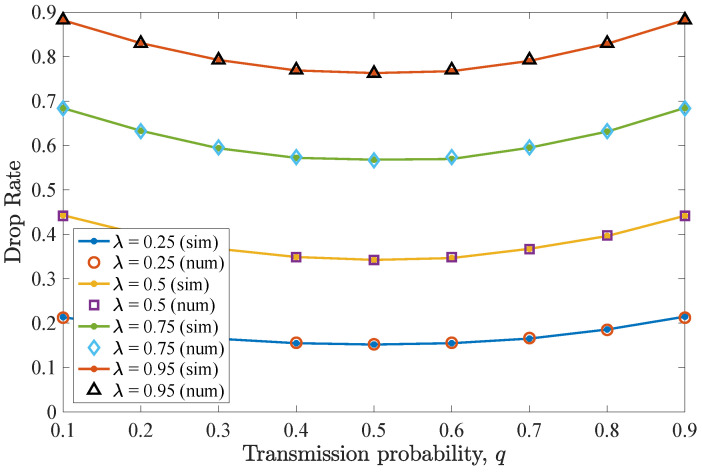
Drop rate for various *q* and λ values for the first DTMC for which n=D−1=2. A perfect match between the numerical and simulation results is observed. Increasing λ results in more packet drops.

**Figure 5 entropy-26-00655-f005:**
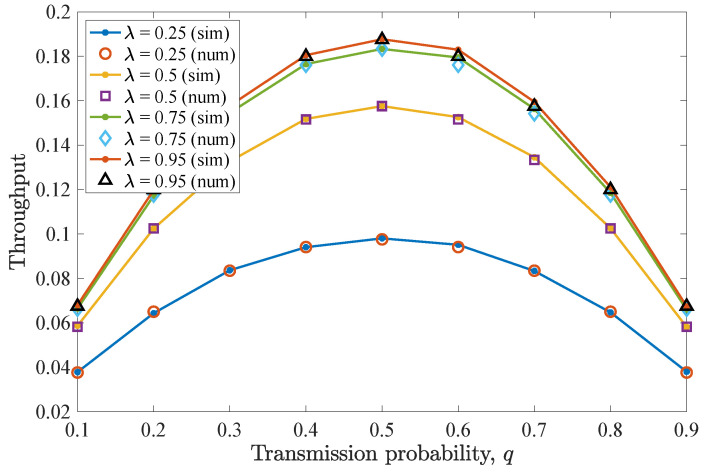
Average throughput for various *q* and λ values for the first DTMC for which n=D−1=2. Throughput is maximized when q=0.5. As λ increases, the throughput increases, and saturation is observed.

**Figure 6 entropy-26-00655-f006:**
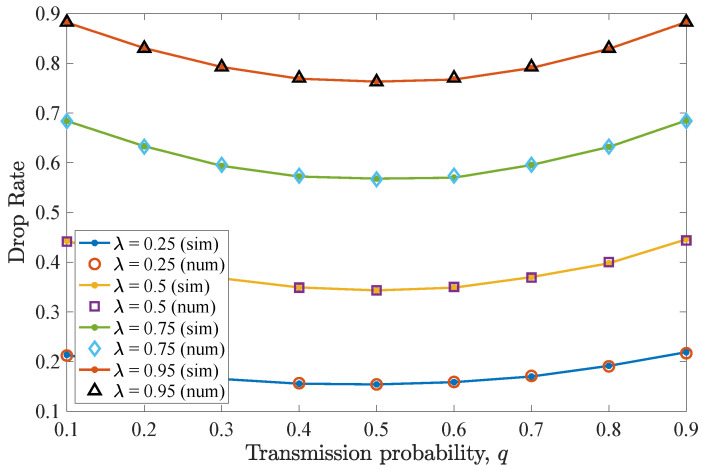
Drop rate for various *q* and λ values for the second DTMC for which D=3 and n=1. Drop rate is slightly increased for higher *q*, compared to the case of n=D−1. More packets are dropped when λ increases.

**Figure 7 entropy-26-00655-f007:**
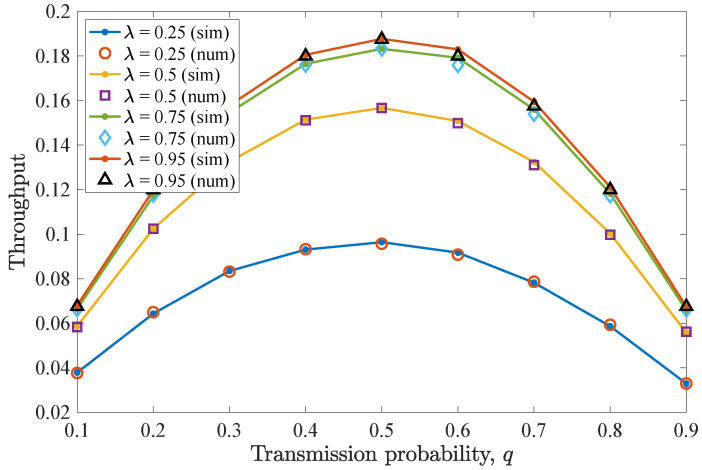
Average throughput for various *q* and λ values for the second DTMC for which D=3 and n=1. Throughput slightly reduces compared to the case of n=D−1 and increases for higher λ values.

**Figure 8 entropy-26-00655-f008:**
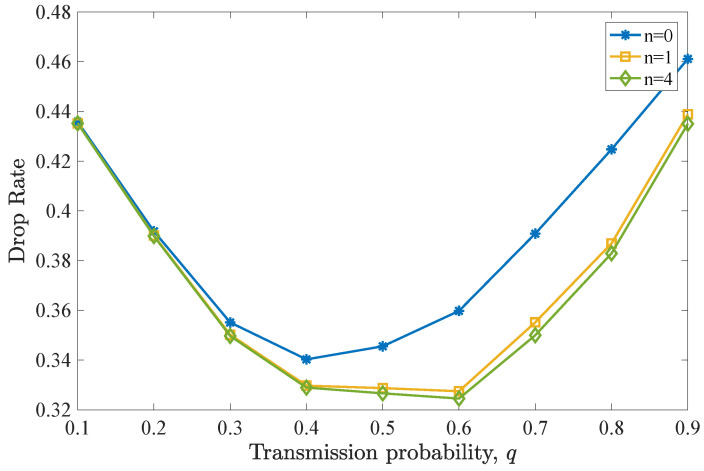
Drop rate for various *q* values for D=5 and n=0,1,4. Drop rate performance improves for increasing *n*. Allowing n=4 retransmissions offers a small drop rate reduction, compared to the case of n=1.

**Figure 9 entropy-26-00655-f009:**
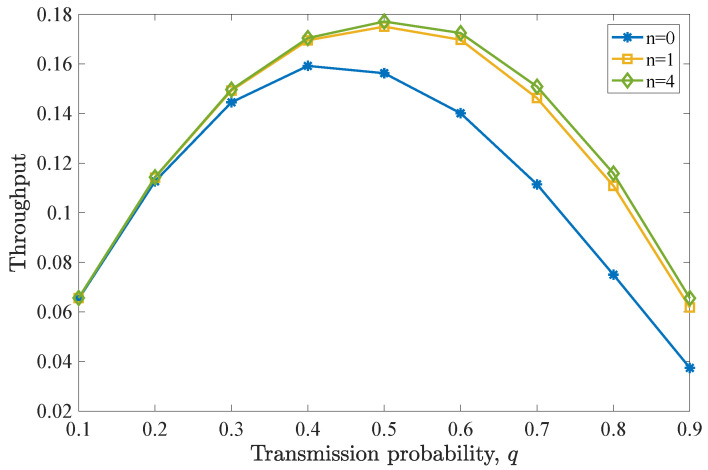
Average throughput for various *q* values for D=5 and n=0,1,4. Throughput is enhanced with increasing *n*. However, only a slight improvement is observed when n=4 is adopted over n=1.

**Figure 10 entropy-26-00655-f010:**
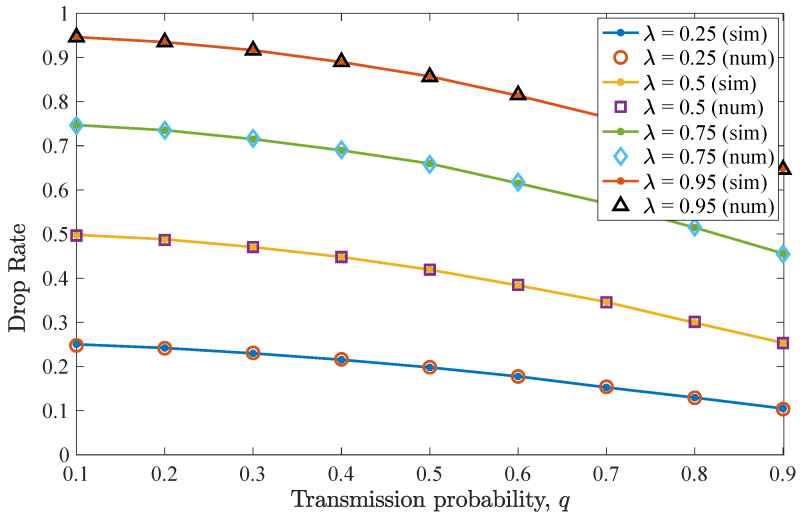
Drop rate for various *q* and λ values for the first DTMC for which n=D−1=2 and N=c=2. It is observed that the drop rate reduces as *q* increases when MPR is allowed. Higher λ values have a negative impact on the drop rate, independently of *q*.

**Figure 11 entropy-26-00655-f011:**
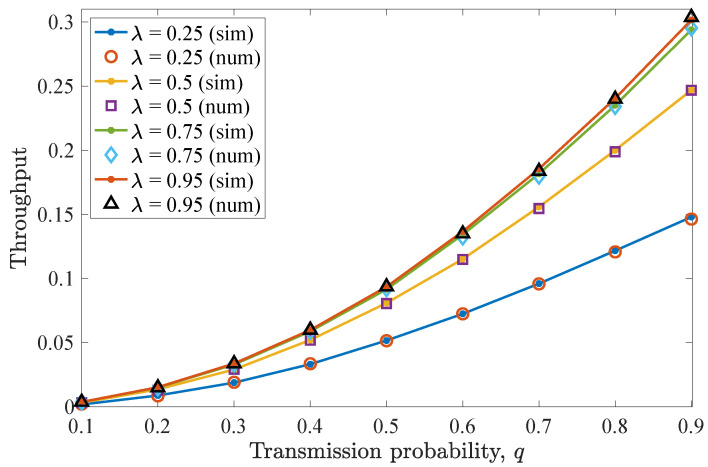
Average throughput for various *q* and λ values for the first DTMC for which n=D−1=2 and N=c=2. As MPR is allowed, increasing *q* has a beneficial impact on throughput. For increased λ values, the throughput is significantly increased for higher *q*.

**Figure 12 entropy-26-00655-f012:**
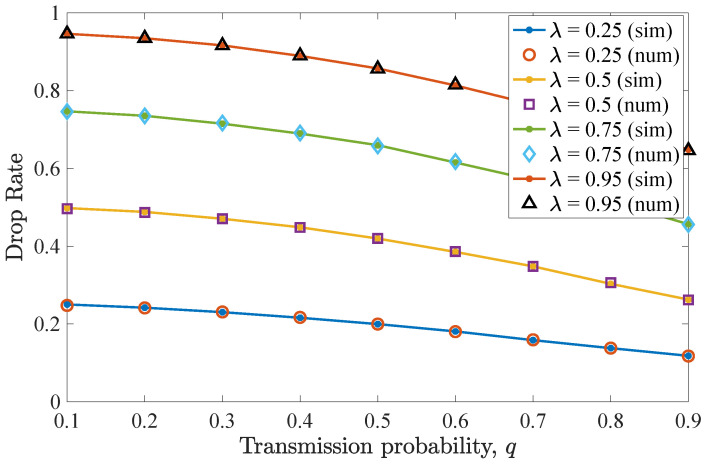
Drop rate for various *q* and λ values for the second DTMC for which D=3, n=1 and N=c=2. Here, the drop rate slightly increases compared to the case of n=D−1. A higher λ degrades the drop rate performance for all *q* values.

**Figure 13 entropy-26-00655-f013:**
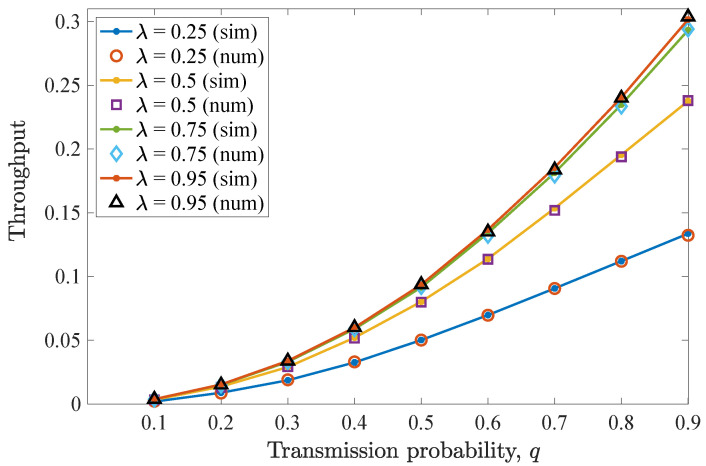
Average throughput for various *q* and λ values for the second DTMC for which D=3, n=1, and N=c=2. The throughput performance is slightly worse compared to the case of n=D−1. As λ increases, the throughput performance improves for larger *q* values.

**Figure 14 entropy-26-00655-f014:**
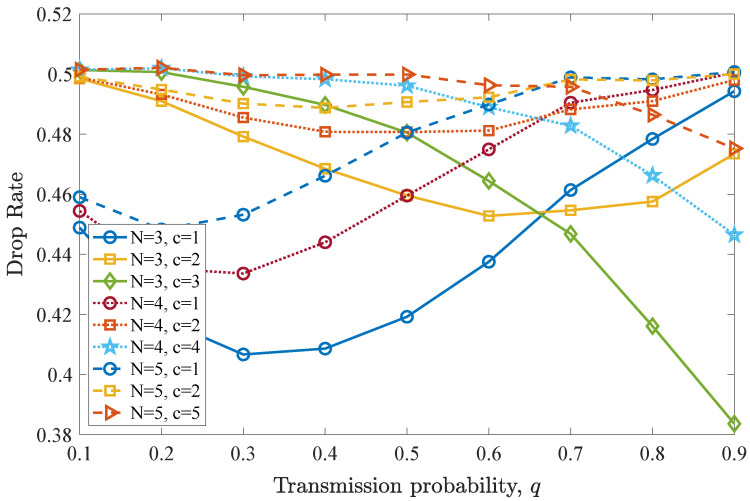
Drop rate for various *q* values for the first DTMC for which n=D−1=2, N=3,4,5, and c=1,⋯,N. A significantly different performance is observed when N>c compared to N=c where the drop rate reduces as *q* increases.

**Figure 15 entropy-26-00655-f015:**
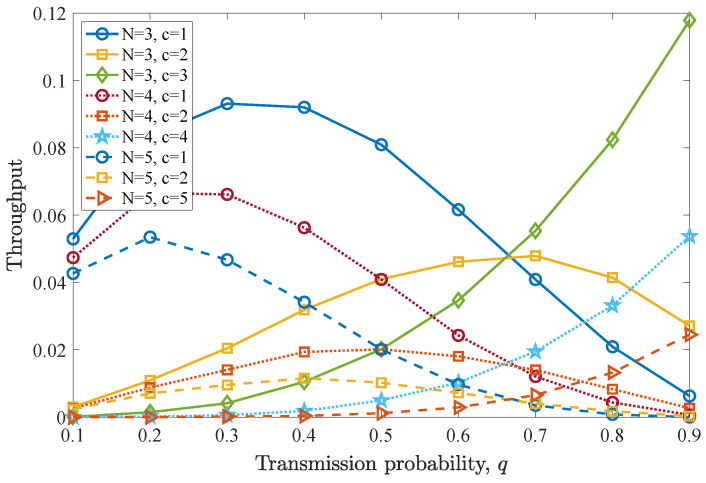
Average throughput for various *q* values for the first DTMC for which n=D−1=2, N=3,4,5, and c=1,⋯,N. Throughput performance for N>c is optimal for different *q* values, while for N=c, an ever-increasing *q* provides better throughput.

**Table 1 entropy-26-00655-t001:** Summary of notation.

Symbol	
*N*	No. of nodes in the network
*D*	Packet deadline
*n*	No. of retransmissions
γ	Signal-to-Interference-and-Noise Ratio (SINR) threshold
$P_{tx}$	Transmit power
$P_{rx}$	Received power
*h*	Small-scale fading random variable (RV)
*s*	Received power factor
α	Path loss exponent
$r_{i}$	Distance between node *i* and the receiver
*v*	Rayleigh fading RV parameter
η	Noise power at the receiver
*q*	Transmission probability
*p*	Success transmission probability
T	Set of transmitting nodes
*c*	No. of transmitting nodes
λ	Average probability of the packet arrival process
μ	Success transmission probability of a packet at the head of the queue
ν	Ratio of *q* over μ
*b*	No. of backlogged nodes
$p_{i,c-1}$	Success transmission probability for node *i* when *c* nodes simultaneously transmit
$p_{s}\left(n,D\right)$	Success transmission probability before packet expiration or before reaching the allowed no. of retransmissions
*S*	Successful transmission event
*X*	First transmission attempt event
*U*	Unsuccessful transmission event
*M*	Transition probability matrix
π(s)	Steady-state of state *s* belonging in the set of states S from which a successful transmission can take place
$\pi (f_{D})$	Steady-state of state $f_{D}$ belonging in the set of states $F_{D}$ transmission is the last
F	The rest of the states from which the last unsuccessful transmission of a packet can take place

**Table 2 entropy-26-00655-t002:** Network parameters.

Parameter	Symbol	Value	Unit
SINR threshold	$\gamma_{i}$	0	dB
Noise power	η	−115.4	dBm
Transmit power	$P_{tx_{i}}$	0.01	W
Rayleigh RV parameter	$v_{i}$	1	
Transmitter–receiver distance	$r_{i}$	100	m
Transmitter–receiver PL exponent	$\alpha_{i}$	4.5	

## Data Availability

The original contributions presented in the study are included in the article, further inquiries can be directed to the corresponding author.

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
