# Peer review of "On the Interplay between Deadline-Constrained Traffic and the Number of Allowed Retransmissions in Random Access Networks"

_entropy, 2024, doi:10.3390/e26080655_

Round 1

Reviewer 1 Report

Comments and Suggestions for Authors

The paper runs to 19 pages in length. The main body of the work includes fifteen figures, fourteen equations and two tables. The title of the work is appropriate, and the abstract provides a suitable summary of the work. There are thirty-three references to work that is relevant.  The choice of keywords is appropriate. The article is divided into five sections, with the simulation and numerical results section being the lengthiest.

The introduction provides a suitable opening to the article. The subject matter is relevant to the special issue and the research community, in general. The derivation of the network and channel models presented in section 2 are clearly articulated in a step-by-step fashion. The paper includes further mathematical derivations, in section 3. Several graphs are used to highlight the significance of the results of the different scenario analyses presented in section 4. The paper closes with a suitable summary, which includes a pointer to future directions in the field of investigation.

The paper is viewed to be of a good standard of presentation and contains material suitable for publication. Due consideration should be given to the following suggestions that may enhance the quality of the presentation:

Line 138:

ARQ is usually denoted as Automatic Repeat Request – Retransmission is currently used in place of Repeat

Line 210, 212 and 237:

Give all equations numbers. The equations associated with the stated lines are currently not numbered.

Lines 235 and 2247:

Expand on the respective paragraphs referring to Figures 1 and 2 to help the reader better understand the significance of the graphs.

Line 288:

Table 2. Expand the Table to four columns to make the presentation easier to read, of the form:

 Parameter       Symbol               Value   Unit

Line 464:

Section 5.2 Future Directions

Replace the bullet list with a paragraph of text.

Author Response

Comment 1: The paper is viewed to be of a good standard of presentation and contains material suitable for publication. Due consideration should be given to the following suggestions that may enhance the quality of the presentation:

Response:  We would like to thank the reviewer for her/his positive asessment. Please find below our answers to all of the reviewer's concerns.

Comment 2:  Line 138:

ARQ is usually denoted as Automatic Repeat Request – Retransmission is currently used in place of Repeat

Response: We have corrected the mistake.

Comment 3: Line 210, 212 and 237:

Give all equations numbers. The equations associated with the stated lines are currently not numbered.

Response: Thank you for highlighting this issue. In this revision, all equations are numbered.

Comment 4: Lines 235 and 2247:

Expand on the respective paragraphs referring to Figures 1 and 2 to help the reader better understand the significance of the graphs.

Response: We have added further discussion on the two figures to highlight their importance and expanded Section 3.2.2.

Comment 5: Line 288:

Table 2. Expand the Table to four columns to make the presentation easier to read, of the form:

 Parameter       Symbol               Value   Unit

Response: We have expanded Table 2 based on your suggestion.

Comment 6: Line 464:

Section 5.2 Future Directions

Replace the bullet list with a paragraph of text.

Response: We agree with your remark. In the revised paper, we have completely re-written Section 5.2.

Reviewer 2 Report

Comments and Suggestions for Authors

This paper presents a solid study and is nearly ready for publication. However, there are a few minor concerns to address:

1. Can the authors comment the construction complexity of the Markov chain for massive multiple access? 

2. In Eq. (1), if the defined quantity denotes power, why is the channel coefficient g_j ​not squared (i.e., g_j^2​)?

3. The quantity q is defined twice in Table 1, which needs clarification.

Comments on the Quality of English Language

This paper is well written. The word "i.e." should be "i.e.," when used.

Author Response

Comment 1: This paper presents a solid study and is nearly ready for publication. However, there are a few minor concerns to address:

Response: We would like to thank the reviewer for the positive evaluation of our work. Please find below our answers to the reviewer's concerns.

Comment 2: Can the authors comment the construction complexity of the Markov chain for massive multiple access? 

Response: We expanded Section 3.2.2 stating that "The construction complexity of the Discrete-Time Markov chain (DTMC) is determined by the deadline $D$ and the number of maximum retransmissions $n$. The number of users only affects the transition probabilities of the DTMC. That is why the DTMC has $D(n+1)+1 -\frac{n(n+1)}{2}$ states, which is independent of the number of users."

Comment 3: In Eq. (1), if the defined quantity denotes power, why is the channel coefficient g_j ​not squared (i.e., g_j^2​)?

Response: Thank you for highlighting this issue. We have fixed this part in the revision.

Comment 4: The quantity q is defined twice in Table 1, which needs clarification.

Response: We have corrected this mistake.

Finally, we have fixed all instances where i.e. appear without a comma.